# Regulation of *VEGFA*, *KRAS*, and *NFE2L2* Oncogenes by MicroRNAs in Head and Neck Cancer

**DOI:** 10.3390/ijms23137483

**Published:** 2022-07-05

**Authors:** Caroline Izak Cuzziol, Ludimila Leite Marzochi, Vitória Scavacini Possebon, Rosa Sayoko Kawasaki-Oyama, Marlon Fraga Mattos, Vilson Serafim Junior, Letícia Antunes Muniz Ferreira, Érika Cristina Pavarino, Márcia Maria Urbanin Castanhole-Nunes, Eny Maria Goloni-Bertollo

**Affiliations:** 1Research Unit of Genetics and Molecular Biology (UPGEM), Department of Molecular Biology, Faculty of Medicine of Sao Jose do Rio Preto (FAMERP), Sao Jose do Rio Preto 15090-000, Brazil; carolcatk@hotmail.com (C.I.C.); ludimila.marzochi@gmail.com (L.L.M.); rosa.oyama@famerp.br (R.S.K.-O.); marlon.fraga.mattos@outlook.com (M.F.M.); leticiaamferreira@gmail.com (L.A.M.F.); erika@famerp.br (É.C.P.); mcastanhole@gmail.com (M.M.U.C.-N.); 2Institute of Biosciences, Humanities and Exact Sciences, Campus Sao Jose do Rio Preto, São Paulo State University (Unesp), Sao Jose do Rio Preto 15054-000, Brazil; vitoria.possebon@unesp.br (V.S.P.); juniorgrolla21@gmail.com (V.S.J.)

**Keywords:** miR-17-5p, miR-140-5p, miR-874-3p, angiogenesis

## Abstract

Mutations and alterations in the expression of *VEGFA*, *KRAS*, and *NFE2L2* oncogenes play a key role in cancer initiation and progression. These genes are enrolled not only in cell proliferation control, but also in angiogenesis, drug resistance, metastasis, and survival of tumor cells. MicroRNAs (miRNAs) are small, non-coding regulatory RNA molecules that can regulate post-transcriptional expression of multiple target genes. We aimed to investigate if miRNAs hsa-miR-17-5p, hsa-miR-140-5p, and hsa-miR-874-3p could interfere in *VEGFA*, *KRAS*, and *NFE2L2* expression in cell lines derived from head and neck cancer (HNC). FADU (pharyngeal cancer) and HN13 (oral cavity cancer) cell lines were transfected with miR-17-5p, miR-140-5p, and miR-874-3p microRNA mimics. RNA and protein expression analyses revealed that miR-17-5p, miR-140-5p and miR-874-3p overexpression led to a downregulation of *VEGFA*, *KRAS*, and *NFE2L2* gene expression in both cell lines analyzed. Taken together, our results provide evidence for the establishment of new biomarkers in the diagnosis and treatment of HNC.

## 1. Introduction

Head and neck cancer (HNC) is a group of cancers that occur in the head and neck region. They represent the seventh most common cancer worldwide and despite advances in treatment, mortality rate has not significantly improved in recent decades. The main risk factors associated with HNC include alcohol consumption and tobacco use [1,2]. Most frequent treatment option include surgery and/or radiation therapy, which can be combined with chemotherapy [3].

During tumor progression, the mechanisms of cellular plasticity and the development of metastasis are coordinated by a complex network of genomic and epigenomic alterations, aspects that represent a major challenge for the diagnosis and treatment of the disease [4]. The range of mutations caused in proto-oncogenes, which consequently lead to changes in the regulation of signaling and cell regulation pathways, are responsible for conferring a high capacity for proliferation, resistance, and survival to tumor cells [5].

Vascular endothelial growth factor (VEGF) is fundamental for tumor angiogenesis [6]. The formation of new blood vessels from pre-existing vessels plays an essential role for tumor development and growth. Indeed, antiangiogenic therapies, such as blocking VEGFA pathway, have been extensively studied [7].

Somatic mutations on the Kirsten Rat Sarcoma Virus (*KRAS*) gene are frequently found in non-small cell lung, pancreas, and colorectal cancers, but overall, approximately 30% of all human cancers display mutation in this important oncogene [8]. Alterations in *KRAS* expression result in uncontrolled cell proliferation and survival, modifications that favor the development of metastases [9]. *KRAS* is an upstream activator of the EGFR Ras/Raf/MEK/ERK signaling pathway, frequently targeted by chemotherapeutic drugs [10]. In addition, activation of *KRAS* is linked to an upregulation of Hypoxia-Induced Factor 1 alpha (*HIF1A*), which in turn can activate *VEGFA* [11,12].

Nuclear Erythroid Factor 2-like-2 (*NFE2L2*/*Nrf2*) is another gene the high expression of which is linked to cancer survival, aggressiveness, and treatment resistance. *NFE2L2* is primarily responsible for the cellular defense mechanism against oxidative and electrophilic stress [13,14]. Treatments targeting *NFE2L2* inhibition have shown promising results in tumors dependent on its activation [15].

MicroRNAs (miRNAs) are small molecules of non-coding RNAs consisting of approximately 18–25 nucleotides that regulate gene expression of several genes [16,17]. MiRNAs regulate gene expression at a post-transcriptional level by binding to the 3′ untranslated regions of target genes [18]. Non-coding regulatory RNAs may play an important role in cancer development and progression [19]. MiRNAs can act as oncogenes or tumor suppressors, influencing many aspects of cancer biology such as proliferation, differentiation, angiogenesis, and metastasis [20,21]. Moreover, some studies have demonstrated that miRNAs are released into the circulation and that the circulating miRNAs are altered during various pathologic conditions, such as inflammation, infection, metabolic disorders, and sepsis [22,23,24].

This study aimed to investigate if the transfection of miR-17-5p, miR-140-5p, and miR-874-3p microRNAs could affect the gene and protein expression of VEGFA, KRAS, and NFE2L2 in two cancer cell lines derived from HNC.

## 2. Results

### 2.1. Bioinformatics-TCGA Analysis

At first, bioinformatics analysis was performed on The Cancer Genome Atlas Program (TCGA) database through the UAL-CAN [25] website. This analysis showed overexpression of the oncogenes *VEGFA* and *NFE2L2*, while *KRAS* was slightly reduced, in tumor samples of HNC (Figure 1).

### 2.2. Expression of miR-17-5p, miR-140-5p, and miR-874-3p in HNC Cell Lines

After transfection, the expression levels of miR-17-5p, miR-140-5p, and miR-874 -3p increased compared to the negative control (RQ = 1.00) in both cell lines (Table 1).

### 2.3. Gene Expression of VEGFA in Transfected Cells

FADU and HN13 cell lines transfected with the mimics miR-17-5p, miR-140-5p, and miR-874-3p showed down expression of *VEGFA* (Figure 2).

### 2.4. KRAS Gene Expression in Transfected Cells

FADU cell line transfected with miR-17-5p, miR-140-5p, and miR-874-3p miRNAs showed decreased KRAS expression. On the other hand, HN13 cell line showed an increase in KRAS expression after miR-17-5p transfection. Conversely, there was a decrease in KRAS expression after microRNAs miR-140-5p and miR-874-3p transfection (Figure 3).

### 2.5. NFE2L2 Gene Expression in Transfected Cells

NFE2L2 expression was downregulated in both cell lines after transfection with miR-17-5p and miR-140-5p. On the other hand, NFE2L2 expression was upregulated in FADU cells transfected with miR-874-3p. The same was not observed for HN13 cell lines, which showed a downregulation of NFE2L2 after miR-874-3p transfection (Figure 4).

Access to The Cancer Genome Atlas (TCGA) database allows for large-scale global gene expression profiling and database mining for potential correlation between genes and miRNAs. The findings on the platform, regarding the expression of these oncogenes in HNC, contributed to the achievement of the objectives of our study. The results indicate that miR-140-5p significantly reduced the expression of VEGFA, NFE2L2, and KRAS genes in both cell lines analyzed. Likewise, miR-17-5p led to a downregulation of VEGFA, NFE2L2, and KRAS expression only in FADU cell line. VEGFA gene expression was also downregulated by miR-874-3p transfection in FADU cell line.

### 2.6. Expression of VEGFA Protein in Transfected Cells

The expression of VEGFA protein was slightly reduced in FADU cells transfected with mimics miR-17-5p, miR-140-5p and miR-874-3p. However, the transfection of the mimics in HN13 cells did not affect VEGFA (Figure 5).

### 2.7. Expression of KRAS Protein in Transfected Cells

Mimics miR-17-5p, miR-140-5p and miR-874-3p transfection induced KRAS protein expression in both HNC cell lines (Figure 5).

### 2.8. Expression of NFE2L2/NRF2 Protein in Transfected Cells

The expression of Nrf2 (NFE2L2) protein was slightly reduced in FADU cells transfected with mimic miR-140-5p, however, miR-17-5p and miR-874-3p left the levels of Nrf2 unaltered in FADU cells. On the other hand, Nrf2 (NFE2L2) protein was upregulated in HN13 cells transfected with the microRNAs (Figure 5).

## 3. Discussion

The present study aimed to evaluate the role of non-coding mRNA (miR-17-5p, miR-140-5p, and miR-874-3p) in the expression of *VEGFA*, *KRAS*, and *NFE2L2* in two different HNC cell lines.

Our results show that *VEGFA* expression was downregulated after miRNA expression, irrespective of the sequence analyzed. *VEGFA* is known for its fundamental role in angiogenesis. It stimulates the generation of new blood vessels from pre-existing vessels, in a mechanism essential for tumor growth and development [26,27]. As previously demonstrated, the expression of miRNAs might significantly influence tumor biology through the regulation of target genes, such as *VEGFA* [28]. Interestingly, the decrease in *VEGFA* expression after miR-17-5p transfection correlates with a previous study on laryngeal cancer, in which miR-17-5p is shown to reduce PI3KR1 expression. PI3KR1 is a subunit of PI3K, one of the signaling pathways linked to *VEGFA* activation [29]. Similar results were found after miR-140-5p transfection in both HNC cell lines. This result is supported by studies reporting that *VEGFA* expression is affected by miR-140-5p expression in other types of cancer. As previously reported, miR-140-5p overexpression suppresses cell proliferation, migration, and invasion processes. It also induces apoptosis of esophagus cancer cells [30]. In addition, miR-140-5p also affects the PI3K/AKT signaling pathway, responsible for coordinating many cellular responses that lead to *VEGFA* upregulation [31,32]. Our results also show that VEGFA expression was downregulated after miR-874-3p transfection in HNC cells. Recently, a study performed on hepatocellular carcinoma showed that inhibition with miR-874-3p did not reduce the expression of the *VEGFA* at the mRNA level, but reduced protein expression at a certain extent [33]. Interestingly, Yuan and collaborators [34] have observed that miR-874-3p overexpression decreased the expression of the signal transducer and activators of transcription 3 (STAT3) mRNA. *STAT3* is known to be part of the signaling cascade controlling *VEGFA* activation. Therefore, it is reasonable to consider that miR-874-3p overexpression could influence *VEGFA* expression both directly and indirectly by targeting other members of *VEGFA* signaling pathway [35].

To the best of our knowledge, no other study has evaluated the association between miRNAs miR-17-5p, miR-140-5p, miR-874-3p and *KRAS* expression. Our results show that miRNA can indeed affect certain aspects of head and neck carcinogenesis. As has long been known, the *KRAS* oncogene can activate or inactivate many signaling pathways, including RAS/RAF/MEK/ERK and PI3K/AKT/mTOR, both linked to *VEGFA* expression, as well as cell proliferation, differentiation, and survival [36]. *KRAS* is also one of the most frequently mutated genes in cancer [37], making it an important therapeutic target [8]. We observed that HNC cell lines from different anatomic sites displayed disparities regarding *KRAS* gene expression upon miR-17-5p overexpression. While FADU cells showed a decrease in *KRAS* expression, HN13 showed an increase in *KRAS* expression. Tumor heterogeneity and microenvironment might be linked to this observation. What is known is that miR-17-5p is commonly associated with the regulation of oncogenes from the PI3K/AKT signaling pathway [38] facilitating even the vascular repair process after aneurysms in a PI3K/AKT/VEGFA pathway dependent manner [39]. As observed for miR-874-3p, it is also possible that miR-17-5p is linked to the regulation of *KRAS* signaling pathway in a way that might even interfere with *VEGFA* expression. On the other hand, our results with miR-140-5p transfections show a decrease in *KRAS* expression, irrespective of the cell line analyzed. This suggests that miR-140-5p is directly involved in *KRAS* expression. In particular, the overexpression of miR-874-3p, although not significant, also exhibits a decrease in *KRAS* expression, which may be explained by the influence of this miRNA on the *VEGFA* expression levels. Therefore, it is possible that miR-874-3p plays an indirect role in *KRAS* expression by *VEGFA* pathway modulation.

*NFE2L2* is one of the main regulatory genes controlling significant cytoprotective effects on oxidative stress through the Nrf2–anti-oxidant response element (ARE) pathway and it is associated with tumor cell survival and treatment resistance [13,40]. It is also referred to as a *VEGFA* expression regulator. Data from colorectal cancer cell lines, human endothelial cells and even zebrafish models have revealed that *NFE2L2* downregulation results in *VEGFA* repression and consequently, angiogenesis inhibition [41,42]. Moreover, mutant *KRAS* transcriptionally promotes metabolic reprogramming and upregulates Nrf2/NFE2L2, and it plays a critical role in anabolic cancer metabolism by altering glucose and glutamine metabolism *KRAS* enhances chemoresistance by upregulating Nrf2 signaling, critical for tumor progression. Therefore, oncogenic *KRAS* enhances chemoresistance by upregulating Nrf2 signaling [43,44]. Our results show that miR-17-5p overexpression is capable of downregulating *NFE2L2* in both cell lines analyzed. It is known that in multiple myeloma cell lines, the overexpression of miR-17-5p in association with Nrf2 influences ferroportin (FPN1) expression, as well as promotes cell proliferation, cell cycle progression and apoptosis inhibition [45]. In thyroid cancer cells, the expression of miR-17-5p was reported to be a potential biomarker, regulating *NFE2L2* expression [46]. We also observed that upon miR-140-5p overexpression, there was a significant reduction in *NFE2L2* expression both in FADU and HN13 cells. Interestingly, upregulation of miR-140-5p is known to promote an increase in oxidative stress and ROS production by suppressing Nrf2 protein expression in a mouse model of atherosclerosis and hypertension [47]. Our results using miR-874-3p reveal that this miRNA promotes an increase in *NFE2L2* gene expression in the FADU cell line and a slight decrease in HN13 cells. This difference might be associated with an indirect effect in other members of the signaling pathway that control *NFE2L2* gene expression. Studies associating *NFE2L2* gene expression and miRNA regulation are scarce, but one study has revealed that the inhibition of miR-144-3p led to an increase in *NFE2L2* expression. The opposite has also been observed, miR-144-3p upregulation induced *NFE2L2* repression in lung cancer cells. Taken together, these results demonstrate the importance of understanding how miRNAs might affect *NFE2L2* expression [48].

The levels of VEGFA protein in FADU cells were completely regulated after transfection with the miRNAs miR-17-5p, miR-140-5p and miR-874-3p. Although our results show significant changes in *VEGFA*, *KRAS*, and *NFE2L2* mRNA expression, protein abundance was only mildly affected. It is known that protein abundance in cells depends on four different events: transcription rates, mRNA half-lives, translation rate and protein half-lives [49,50]. The imperfect correlation between protein and mRNA levels can be explained by technical and/or biological issues [51]. The relevance for the phenotype is also a matter of intense debate. Still, understanding how mRNA levels translate into protein activity and its role in cancer progression is undoubtedly important.

The *VEGFA*, *KRAS*, and *NFE2L2* genes are involved in relevant biological processes related to tumorigenesis (Figure 6). Our results show that miR-17-5p, miR-140-5p, and miR-874-3p might work as important regulators of *VEGFA*, *KRAS*, and *NFE2L2* signaling pathways. This suggests a possible use of these miRNAs as therapeutic targets for HNC treatment. Further studies, and an in vivo work, are needed to confirm that these miRNAs can act as biomarkers for HNC.

## 4. Materials and Methods

### 4.1. Cell Lines Culture

HNC cancer cell lines FADU (pharyngeal cancer) and HN13 (cancer of the oral cavity) were thawed and cultured in 25 cm^2^ and 75 cm^2^ flasks containing high glucose DMEM culture medium (Sigma, San Louis, MO, USA), supplemented with 10% fetal bovine serum (Gibco, Grand Island, NY, USA), 100 units/mL of sodium penicillin, 100 µg/mL of streptomycin (Invitrogen, Waltham, MA, USA) and 1% L-glutamine (Gibco, Grand Island, NY, USA). Flasks were kept at 37 °C in an 5% CO^2^ atmosphere. Media was replaced every three days. Cells were trypsinized when 80% confluence was reached, and the number of cells needed to carry out the transfection was obtained in the second passage for the FADU cell line and in the first passage for the HN13 cell line. Shortly after, cells were washed twice with 1x PBS and treated with Trypsin/EDTA (0.125%/0.05%). Trypsinization was interrupted by the addition of Complete Culture Medium. Cells were then split into new flasks to be kept or to be used in further experiments.

### 4.2. Transfection of miRNAs in Cell Lines

The miRNAs miR-17-5p, miR-140-5p, and miR-874-3p were predicted for the *VEGFA*, *KRAS,* and *NFE2L2* genes using the mirDIP [51,52] and TarBase V.8 [53] platforms.

Indirect transfection of FADU and HN13 cells were performed in 24-well plates containing approximately 80,000 cells, 500 µL of antibiotic-free DMEM medium, 100 µL of Opti-MEM (Invitrogen), 10 mM of mirVana™ miRNA Mimic Negative Control ( Thermo Scientific, Waltham, MA, USA) or the mimics “mirVana^®^ miRNA mimic hsa-miR-17-5p (MC12412 Thermo Scientific, Waltham, MA, USA)” or “mirVana^®^ miRNA mimic hsa-miR-140-5p (MC10205, Thermo Scientific, Waltham, MA, USA) or “mirVana^®^ miRNA mimic hsa-miR-874-3p (MC12355 Thermo Scientific, Waltham, MA, USA)” and 1μL Lipofectamine RNAiMax (Invitrogen, Waltham, MA, USA). Plates were incubated at 37 °C in a 5% CO_2_ atmosphere for 48 h. After the incubation time, cells were harvested for RNA and protein extraction. Gene expression, and miRNA and protein analyses were done afterwards. Three independent experiments were performed following the same experimental conditions.

### 4.3. Gene and miRNA Expression

RNA was extracted from transfected cells with Trizol (Invitrogen) according to the manufacturer’s instructions. RNA samples were quantified using the NanoDrop 2000 (Thermo Fisher Scientific, Waltham, MA, USA). cDNA was obtained using 20 µL reaction containing 0.5–1 µg of total RNA with High Capacity cDNA Archive kit (Life Technologies, Carlsbad, CA, USA), according to the manufacturer’s instructions. To convert the miRNAs into cDNA, TaqMan-Micro RNA Reverse Transcription kit (Applied Biosystems, Waltham, MA, USA) was used. Analyses of tumor cells gene expression and miRNA were performed in duplicate. Real-time PCR was performed to quantify the expression of the proposed genes and microRNAs, using TaqMan MGB probes linked to the FAM fluorophore (Applied Biosystems, Waltham, MA, USA), following the manufacturer’s instructions. Housekeeping genes GAPDH (HS9999905_m1) and RPLPO (HS00420895_g1) were evaluated to normalize the expression of *VEGFA* (HS00900055_m1), *KRAS* (HS00364284_g1), and *NFE2L2* (HS00975961_g1) genes. Endogenous controls U6 (Id: 1973) and RNU48 (Id: 1006) were used to normalize the expression of the miRNAs has-miR-17-5p (Id: 2308), has-miR-140-5p (Id: 1187) and has-874-3p (Id: 2268).

### 4.4. Protein Quantification and Expression

Total protein was isolated from cultured cells using RIPA buffer (Sigma Aldrich, San Louis, USA), according to manufacturer’s instructions. Protein samples were quantified using the BCA Protein Assay Kit (Thermo Fisher Scientific, Waltham, MA, USA). Subsequently, samples were submitted to PAGE followed by Western Blotting (WB). VEGFA, KRAS, and Nrf2 (NFE2L2) proteins were detected using primary antibodies against VEGF (Invitrogen, Waltham, MA, USA MA5-13182), KRAS (Abnova, São Caetano do Sul, Brazil H00003845-M05), and Nrf2 (Invitrogen, Waltham, MA, USA PA5-27882), the three antibodies, were used at 1:1000 dilution, in Secondary antibodies (anti-mouse Sigma A9044-VEGFA and KRAS; anti-rabbit Abcam, Cambridge, UK ab97051-Nrf2) were used at 1:25000 dilution.

### 4.5. Statistical Analysis

Statistical analyses were performed using the GraphPad Prism software, version 9.0. The distribution of continuous data was evaluated using the Shapiro–Wilk normality test. Wilcoxon’s signed classification test and the t-test were used to assess gene expression data. The correlation between the expression of miRNAs and gene expression was analyzed by Spearman’s Correlation. Values of *p* ≤ 0.05 (*) were considered significant.

## 5. Conclusions

Our results demonstrate that the overexpression of miR-17-5p, miR-140-5p, and miR-874-3p miRNAs negatively regulates the expression of *VEGFA*, *KRAS*, and *NFE2L2* genes in HNC cell lines. Many biological processes can be affected by these alterations such as angiogenesis as well as metastasis. Moreover, the cell lines transfected with miR-17-5p, miR-140-5p, and miR-874-3p clearly downregulate the expression of VEGFA at both protein and mRNA levels. More studies are necessary to understand the influence of these miRNAs on the expression of *VEGFA*, *KRAS*, and *NFE2L2* genes. Their role in development and cancer progression and their importance as biomarkers for diagnosis and treatment of HNC should be further explored.

## Figures and Tables

**Figure 1 ijms-23-07483-f001:**
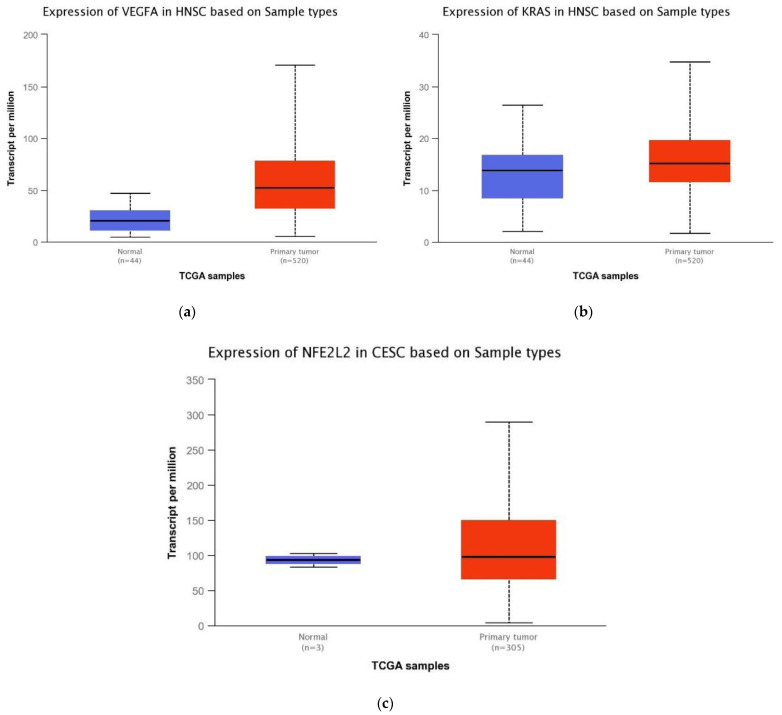
TCGA expression data of (**a**) *VEGFA* (*p* = <0.001), (**b**) *KRAS* (*p* = 0.80) and (**c**) NFE2L2 (*p* = 0.70) in HNC, comparing normal tissue and primary tumor.

**Figure 2 ijms-23-07483-f002:**
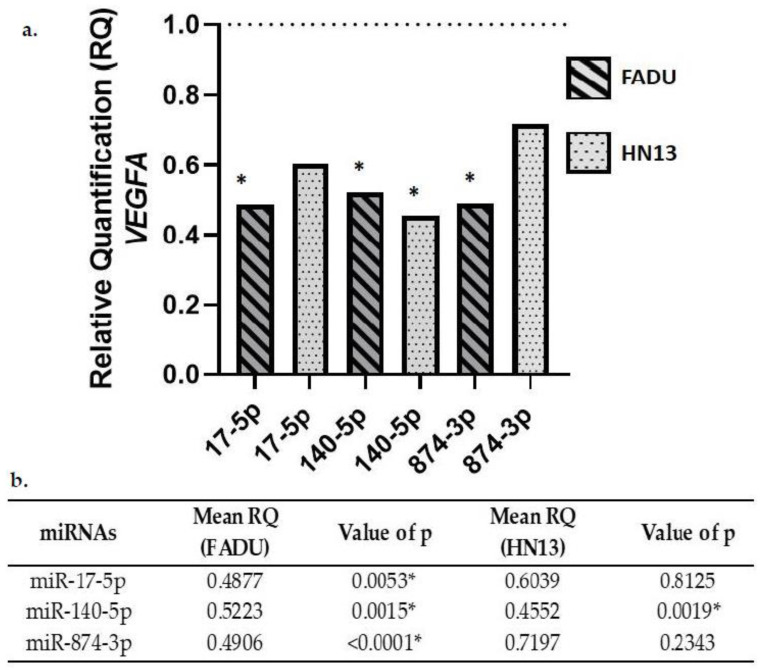
(**a**). *VEGFA* gene expression after miR-17-5p, miR-140-5p and miR-874-3p transfection compared to the negative control (RQ = 1). (**b**) Mean relative quantification (RQ) of *VEGFA* gene expression in HNC cell lines transfected with miR-17-5p, miR-140-5p, and miR-874-3p. * Significant values.

**Figure 3 ijms-23-07483-f003:**
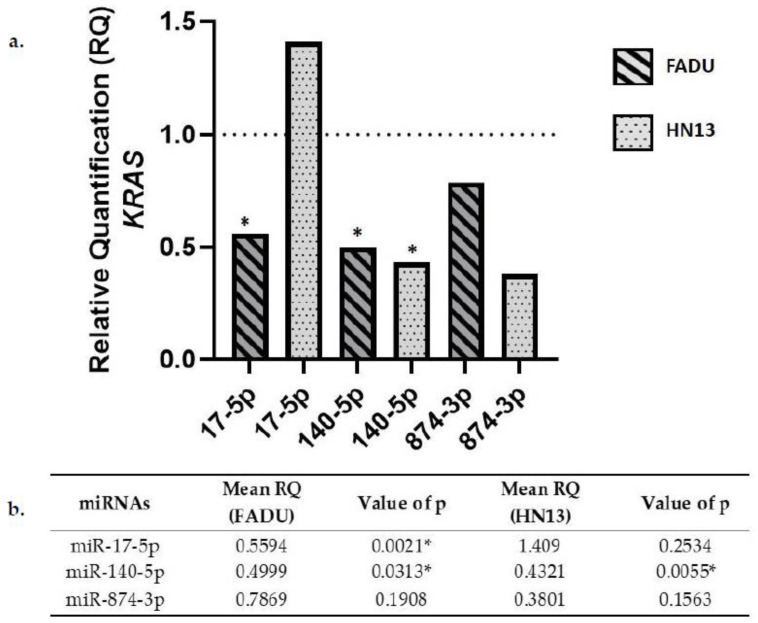
(**a**) KRAS gene expression after miR-17-5p, miR-140-5p and miR-874-3p transfection compared to the negative control (RQ = 1). (**b**) Mean relative quantification (RQ) of KRAS gene expression in HNC cell lines transfected with miR-17-5p, miR-140-5p, and miR-874-3p. * Significant values.

**Figure 4 ijms-23-07483-f004:**
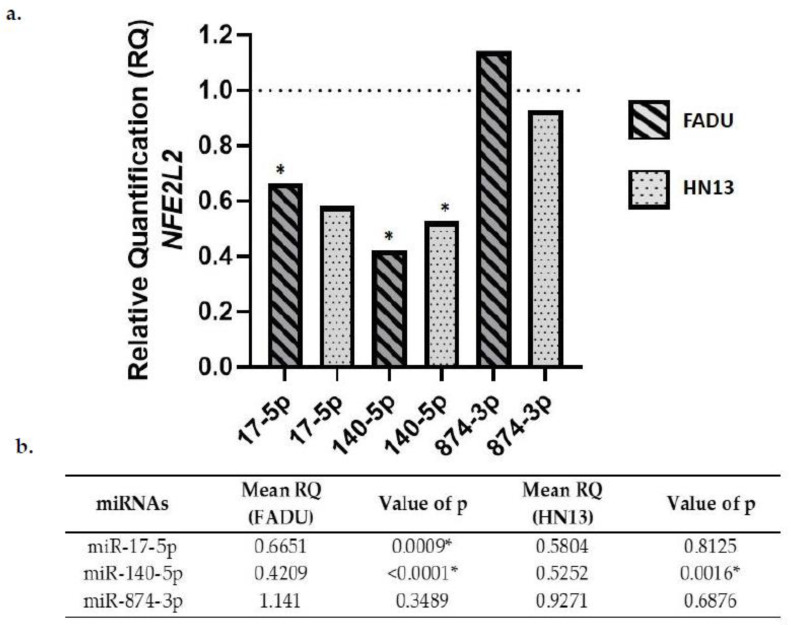
(**a**) *NFE2L2* gene expression after miR-17-5p, miR-140-5p, and miR-874-3p transfection compared to the negative control (RQ = 1). (**b**) Relative quantification mean (RQ) of NFE2L2 gene expression in HNC cell lines transfected with miR-17-5p, miR-140-5p, and miR-874-3p. * Significant values.

**Figure 5 ijms-23-07483-f005:**
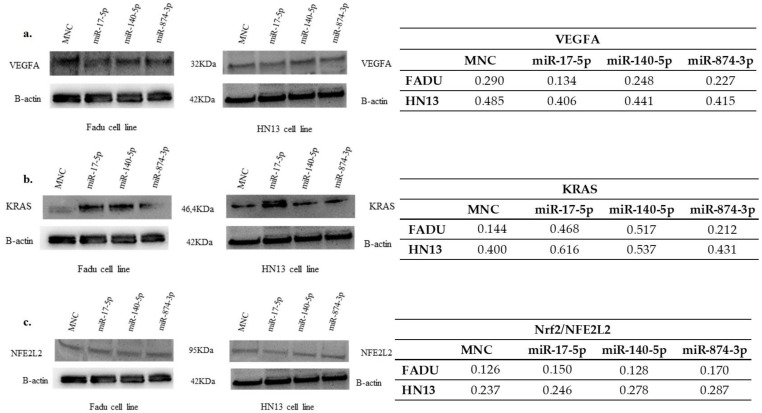
(**a**) VEGFA protein expression in HNC cancer cell lines expressing miRNAs. (**b**) KRAS protein expression in HNC cancer cell lines expressing miRNAs. (**c**) NFE2L2 protein expression in HNC cancer cell lines expressing miRNAs. Abbreviation: MNC, mimic negative control. For the evaluation of protein expression, the images were analyzed and quantified using ImageJ 4.0 software.

**Figure 6 ijms-23-07483-f006:**
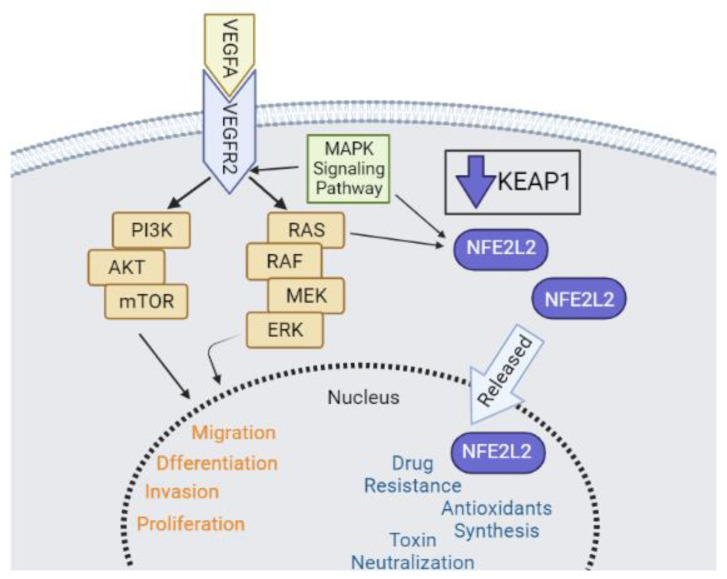
Pathways of action of the *VEGFA*, *KRAS,* and *NFE2L2* genes. VEGFA binds to its membrane receptor VEGFR2 leading to activation of the signaling cascade RAS/RAF. NFE2L2 is regulated by the KEAP1 protein which, when inhibited, increases the levels of NEF2L2 that is released into the cell nucleus. The MAPK pathway is also related to the VEGF, KRAS and NFE2L2 pathway. Adapted: Cuzziol et al., 2020 [35].

**Table 1 ijms-23-07483-t001:** Mean relative quantification (RQ) of microRNAs miR-17-5p, miR-140-5p, and miR-874-3p after transfection in HNC cell lines.

miRNAs	Mean RQ (FADU)	Value of *p*	Mean RQ (HN13)	Value of *p*
miR-17-5p	407.9	0.0087 *	182.2	0.3316
miR-140-5p	15.620	0.0121 *	3.322	0.0024 *
miR-874-3p	2.871	0.0313 *	13.265	0.0642

* Significant values.

## Data Availability

The data can be found at the Research Unit in Genetics and Molecular Biology (UPGEM), at the Faculty of Medicine of Sao Jose do Rio Preto (FAMERP).

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
