# Peer review of "Regulation of VEGFA, KRAS, and NFE2L2 Oncogenes by MicroRNAs in Head and Neck Cancer"

_ijms, 2022, doi:10.3390/ijms23137483_

Round 1

Reviewer 1 Report

All concerns have been addressed, ready for acceptance. 

Author Response

Thank you for accepting our article

Reviewer 2 Report

In this study authors investigated if miRNAs hsa-miR-17-5p, hsa-miR-140-5p, and hsa-miR-874-3p could interfere in VEGFA, KRAS, and NFE2L2 expression in HNC cell lines and found that miR-17-5p, miR-140-5p and miR-874-3p overexpression led to a downregulation of VEGFA, KRAS, and NFE2L2 gene expression in both cell lines analyzed. 

This is a very interesting study but it presents some flaws that must be resolved. In particular: 

Lines 63-66: It deserves to be specified that miRNA are also important factors and markers of non-cancerous diseases such as Gestational diseases (PMID: 32726711, 35615626), neurological disorders (PMID: 35655754, 35602517 ) and metabolic disorders (PMID: 35351622). this is an important point to highlight because VEGFA and NFE2L2 (analysed by the authors in this study) are also impaired in some of these pathologies further validating the results obtained by the authors. 

Figure 1: Please improve image quality

Figure 2 and table 2 should be put in the same figure. The same for  Figure 3 and Table 3, Figure 4 and Table 4 because tables are just the quantification of the instogram reported. In all these figures negative control is missing but must be reported.

Figure 5,6 and 7: Densitometric analysis must be shown. Moreover, It would be helpful reducing the number of figures pulling Western blots in the same figure  with respective densitometric analysis 

Discussion: It deserves to be pointed out that mutations of KRAS (a common oncogenes), lead to an increase in NFE2L2 transcription and activity in cancer cells protecting tumour cells from cytotoxic effects induced by chemotherapeutic drugs, explaning the key role of NFE2L2 in chemoresistance onset.

4.4. Protein quantification and expression: Primary antibodies dilutions must be reported

Author Response

Thank you for the suggestions to improve the quality of our work.

Round 2

Reviewer 2 Report

Manuscript has been significantly improved and can be accepted in the present form. 

Author Response

Thank you for all the considerations and improvements suggested for the article, and for the acceptance for publication.

This manuscript is a resubmission of an earlier submission. The following is a list of the peer review reports and author responses from that submission.

Round 1

Reviewer 1 Report

This study analyzes the effect of microRNAs on the expression of VEGFA, KRAS, and NFE2L2 oncogenes in head and neck cancer cells. The following comments should be taken into consideration before acceptance for publication.

  1. In the Introduction, the authors did not provide clear rationale or evidence leading to the study of regulation of VEGFA, KRAS, and NFE2L2 by specific microRNAs in head and neck cancer.
  2. No functional studies were provided in this study.
  3. No in vivo nor clinical studies were provided in this paper.

Author Response

Dear Reviewer,

Please see the attachment. All suggestions were considered. I await release to submit the article containing the corrections.

Regarding the Editor's comments, the following considerations follow:

Academic Editor Comments: The experiments should be performed at least on three head and neck cancer (HNC) cell lines. A study of HNC patients array should be considered to make any observation/discussion on HNC patients. It is not clear whethere a densitometric analysis of the bands have been performed.

 Answer: One bioinformatic analysis was performed before starting this study on The Cancer Genome Atlas (TCGA) database which shows large-scale global gene expression profiling. Therefore, two cell lines were used to perform the in vitro representation of the regulation of VEGFA, KRAS and NFE2L2 oncogenes by miRNAs 17-5p, 140-5p and 874-3p. We emphasize that the cell lines are from different regions of the HNC, which allows covering the particularities of the tumor microenvironments

            Analysis of the results obtained through the WB technique and quantified in the ImageJ software:

Table 1. VEGF protein levels after transfection of miR-17-5p, miR-140-5p and miR-874-3p miRNAs, were obtained through analysis in ImageJ software.

VEGFA

                               MNC     miR-17-5p     miR-140-5p      miR-874-3p        FADU                    0,290          0,134                0,248                 0,227

                               MNC     miR-17-5p    miR-140-5p      miR-874-3p

HN13                     0,485          0,406                0,441                  0,415

Table 2. KRAS protein levels after transfection of miR-17-5p, miR-140-5p and miR-874-3p miRNAs, were obtained through analysis in ImageJ software.

KRAS

                               MNC     miR-17-5p     miR-140-5p      miR-874-3p        FADU                     0,144           0,468                 0,517                  0,212

                               MNC     miR-17-5p    miR-140-5p      miR-874-3p

HN13                      0,400            0,616               0,537                   0,431

Table 3. NRF2 protein levels after transfection of miR-17-5p, miR-140-5p and miR-874-3p miRNAs, were obtained through analysis in ImageJ software.

Nrf2/NFE2L2

                               MNC     miR-17-5p     miR-140-5p      miR-874-3p        FADU                    0,126            0,150                  0,128                  0,170

                               MNC     miR-17-5p    miR-140-5p      miR-874-3p

HN13                       0,237             0,246                0,278                   0,287

Reviewer 2 Report

Article by Dr. Goloni-Bertollo and group elaborates on the role of VEGFA, KRAS, and NFE2L2 oncogenes in head neck cancer. Though their findings are interesting, several things are required at this moment to be added to this manuscript before it is ready for acceptance. They are as follows:

  1. Authors should discuss whether KRAS activated NRF2 pathways in head neck cancers are similar to what has been mentioned it been shown in pancreatic cancer (PMID: 31911550 and PMID: 21734707). Authors can add a few lines on this aspect in the discussion part.
  2. Authors should also mention whether there is any clinical trial ongoing based on biomarkers such as KRAS, and NFE2L2 mutations in head neck cancers.
  3. Please refer to PMID: 22703241 for the correct molecular weight of NRF2/NFE2L2 in the western blot and repeat the WB with the correct antibody and correct molecular weight in this manuscript.
  4. There should be any mechanism and functional outcome explaining the phenomenon seen in this manuscript. Authors should perform a couple of experiments on this. At least the authors should propose some of them as future experiments in the discussion part as an aspect of future follow-up study.   

Author Response

Dear Reviewer,

Please see the attachment. All suggestions were considered. I await release to submit the article containing the corrections.

Round 2

Reviewer 1 Report

  1. The authors did not respond to my concern that no functional or mechanistic studies are included in this work.
  2. I am not sure Fig. 8 is appropriate or not for an oruginal article since it is adapted from the authors' previous work, not concluded from the current study.

Reviewer 2 Report

The authors addressed all concerns mentioned before. One minor mistake needs to be addressed-

In figure 6 KRAS molecular weight has been marked as 46.4 kDa. It will be 21 kDa. Once that is corrected it should be ready for acceptance. 

Round 3

Reviewer 1 Report

I feel sorry for the funding issue raised by the authors. However, I still stick to my concerns which were not answered satisfactory by the authors.